# Understanding the Intricacies of Khat-Associated Cardiovascular Disease

**DOI:** 10.3390/jcm14041100

**Published:** 2025-02-09

**Authors:** Azka Naeem, Divya Situt, Vijay Shetty, Jacob Shani, Emmanuel U. Emeasoba

**Affiliations:** 1Department of Internal Medicine, Maimonides Medical Centre, Brooklyn, NY 11219, USA; vshetty@maimonidesmed.org; 2School of Medicine, St. George’s University Grenada, True Blue, West Indies, Grenada; divya.situt@gmail.com; 3Department of Cardiology, Maimonides Medical Centre, Brooklyn, NY 11219, USA; jshani@maimonidesmed.org (J.S.); emmanuelemeasoba@gmail.com (E.U.E.)

**Keywords:** khat, amphetamines, cardiomyopathy, hypertension, tachycardia, acute coronary syndrome, immigrant population

## Abstract

Khat, derived from the Catha edulis plant, is widely consumed in East Africa and the Arabian Peninsula, where it holds significant socio-cultural importance. This review examines the multifaceted effects of khat, particularly focusing on its cardiovascular implications. Khat’s active constituents, notably cathinone and cathine, exhibit stimulant and psychoactive properties akin to amphetamines, leading to heightened alertness and euphoria. However, chronic consumption is associated with adverse effects, including cardiovascular diseases such as hypertension, myocardial infarction, and cardiomyopathy. The review highlights the pharmacokinetics of khat, with cathinone being rapidly absorbed and leading to sympathomimetic effects. Studies indicate a correlation between chronic khat use and increased risks of hypertension, acute coronary syndromes, and cardiomyopathy. These cardiovascular conditions are exacerbated by prolonged hemodynamic stress, catecholamine release, and oxidative stress induced by khat’s active compounds. Additionally, khat’s impact extends beyond the cardiovascular system, affecting neurological, reproductive, and gastrointestinal health. Despite its legality in certain regions, khat is classified as a controlled substance in many countries, emphasizing the need for global awareness of its health risks. This review calls for longitudinal studies to elucidate the pathophysiological mechanisms of khat-induced cardiomyopathy and to identify potential biomarkers for its early detection. Furthermore, it advocates for culturally sensitive public health initiatives and clinical guidelines to mitigate the adverse health effects of khat consumption, especially among immigrant populations in developed nations. Recognizing and addressing khat’s cardiovascular implications is crucial for improving patient outcomes and guiding effective clinical practice.

## 1. Introduction and Background on Khat

The consumption of khat leaves, derived from the plant Catha edulis, is a prevalent practice in East Africa and various countries across the Arabian Peninsula. This customary habit involves chewing fresh khat leaves and twigs for approximately 4–6 hours daily. The process entails placing the leaves on one side of the mouth and gradually chewing them to extract the juice, which is then swallowed, while the leaves persist in the buccal cavity. 

Fresh khat contains cathinone, cathine, and norephedrine, along with additional phenylalkylamine alkaloids like phenylpentenylamines, merucathinone, pseudomerucathine, merucathine, and cathedulin alkaloids as well, albeit in relatively low concentrations. These compounds bear resemblance to both amphetamine and noradrenaline. Historically, khat has been utilized to combat depression and fatigue, and is also an appetite suppressant for obese individuals. It has also been employed to alleviate asthma and other lung conditions, colds, fevers, coughs, and headaches. However, the consumption of khat also poses risks. It has the potential to induce alterations in mood; heighten alertness, hyperactivity, and anxiety; increase blood pressure; and cause cardiovascular ailments.

In Ethiopia and southwestern Arabian regions, chewing khat holds profound socio-cultural importance, often serving as a daily recreational activity due to its euphoric, stimulant, and libido effects [1]. Over the last thirty years, its accessibility and utilization have expanded globally, including regions such as the United States and Europe. Predominantly, consumers in Western countries consist of immigrant groups hailing from Eastern Africa or the Middle East, and khat chewing has garnered limited medical attention until recently. Despite this, comprehensive clinical studies on its effects remain limited.

Khat use has deep cultural and historical significance, but its increasing global prevalence raises concerns about its health impact. As khat becomes more accessible beyond traditional regions, understanding its pharmacological and long-term health risks is essential for medical professionals’ and public health awareness.

## 2. The Adverse Effects of Khat

Khat, originating from various geographical regions, encompasses approximately 44 different types, each exhibiting slightly different chemical profiles influenced by environmental and climatic conditions. The core constituents of khat include alkaloids, terpenoids, flavonoids, sterols, glycosides, tannins, amino acids, vitamins, minerals, and cathedulins. Of particular interest are cathinone and cathine due to their stimulating and psychoactive effects [2]. Studies identify various cathedulins present in fresh khat leaves including cathinone (S-(-)-cathinone), cathine (1S, 2S-(+)-norpseudoephedrine or (+)-norpseudoephedrine, and norephedrine (1R, 2S-(-)-norephedrine), which share structural similarities with amphetamine and noradrenaline [3].

The (-) enantiomer of cathinone, found in young leaves at relatively higher concentrations, is more potent and shares the same absolute configuration as S- (+)-amphetamine. However, as the plant matures or is harvested, cathinone is converted into cathine and (-)-norephedrine, which have relatively fewer stimulant effects [4]. Cathinone facilitates the release of catecholamines from sympathetic nerve terminals and inhibits monoamine reuptake, leading to higher synaptic monoamine levels [5]. Cathine has similar but milder effects. Together, cathinone and cathine result in heightened arousal, alertness, and euphoria.

Prolonged or excessive khat consumption can lead to adverse effects across multiple organ systems, exacerbating pre-existing conditions or predispositions in susceptible individuals. A single khat chewing session can involve 100–500 g of khat, with 80% of cathinone and cathine and over 90% of norephedrine extracted in the saliva within 15–45 min [6]. Absorption is primarily initiated in the buccal mucosa, followed by the stomach and small intestines once the juices are swallowed, with effects manifesting within 15–30 min and lasting approximately 3 hours [7]. Notably, khat chewing sessions can extend for 3–7 hours despite the relatively short duration of cathinone’s effects [8].

Khat’s stimulating effects come from its psychoactive components, cathinone and cathine; however, prolonged or excessive use may carry significant health risks. Recognizing these effects is essential for understanding its broader role both medicinally and recreationally.

## 3. Medicinal and Non-Medicinal Uses of Khat

The chewing of khat leaves in some cultures serves multiple purposes, including increasing alertness, relieving fatigue, and enhancing cognitive function. Research conducted by Lemieux et al. has demonstrated that khat consumption leads to appetite suppression in both animals and humans [9]. This effect is primarily attributed to cathinone, the principal psychoactive agent in khat, which induces a stimulant effect along with appetite suppression. Cathinone is linked to the elevation of plasma leptin levels, resulting in reduced hunger and contributing to weight and lipid reduction, independently of ghrelin and Peptide YY secretion [10].

Traditionally, khat has been utilized as a natural remedy for various types of pain, such as headaches, toothaches, and muscle aches. Its analgesic properties are believed to stem from its ability to modulate neurotransmitter levels in the brain. Research by John Connor et al. suggests that khat extract exhibits analgesic effects comparable to D-amphetamine and ibuprofen, indicating its potential as a pain-relieving agent [11]. In certain cultures, the practice of chewing khat is deeply ingrained in social and cultural traditions. It is frequently consumed during social gatherings, ceremonies, and celebrations to facilitate social cohesion and relaxation. Additionally, khat consumption leads to mood elevation and euphoria, contributing to its popularity as a recreational drug in specific cultural contexts.

While khat has a long history of use, its health effects remain a growing concern. Assessing the role of khat would require weighing its perceived benefits against the risk of long-term effects, considering both context and extent.

## 4. Legal Status of Khat

Cathinone and cathine are substances classified as Schedule I and Schedule III, respectively, by both the World Health Organization (WHO) and the Controlled Substance Act. Although khat itself is not explicitly listed as a controlled substance, its detectable levels of cathinone justify its consideration as Schedule I. Despite this classification, khat remains legal and easily accessible in countries such as Yemen, Ethiopia, Somalia, and certain parts of East Africa, although religious institutions in some areas may impose restrictions on its use.

Remarkably, these countries lack specific age restrictions for khat consumption, contributing to a troubling increase in usage among children and teenagers. Notably, khat has emerged as Ethiopia’s primary export commodity after coffee, with a significant expansion in the land allocated to its production over the past two decades. In contrast, khat is strictly prohibited in Canada, the United States, the UK, Germany, France, and various other European countries. However, global migration and advancements in transportation have facilitated the spread of the khat chewing habit worldwide.

## 5. Cardiovascular Effects of Khat

The proportion of premature deaths attributed to cardiovascular diseases (CVDs) displays a significant variance, ranging from 4% in high-income nations to as high as 42% in low-income countries. Notably, there has been an upward trend in CVD-related numbers within high-income countries, which can be linked to the increasing influx of immigrants relocating to these nations. While traditional risk factors associated with CVDs have been identified, research related to specific substances, such as khat chewing, has been either inadequately conducted or lacked effective interventions.

Khat contains several psychoactive compounds, including cathinone and cathine, which exert sympathomimetic effects by enhancing the release of catecholamines such as adrenaline and noradrenaline. This physiological response leads to increased heart rate, blood pressure, and myocardial oxygen demand (Figure 1). Research conducted in Yemen has revealed a correlation between elevated blood pressure (BP) and heart rate (HR) in individuals who chew khat, coinciding with increased plasma cathinone concentrations. Moreover, chronic khat use has been linked to an increased risk of hypertension, diabetes mellitus, and dyslipidemia.

In a study conducted by Getahun W et al., researchers discovered a notable increase in the occurrence of hypertension or the use of antihypertensive medications among khat chewers (13.4%) compared to non-chewers (10.7%). The adjusted odds ratio (AOR) was calculated to be 1.66, with a 95% confidence interval of 1.05–3.13 [12]. This study’s large sample size, comparative design, which accounted for key demographic factors, and its high response rate strengthen its validity. However, the reliance on self-reported data introduces the potential for recall bias, and its geographic focus on Butajira, Ethiopia, limits generalizability. Furthermore, Fikru et al. (2008) reported a significant association between regular khat chewing and elevated mean diastolic blood pressure (β = 1.9, *p* = 0.02) [13]. Prolonged hemodynamic stress on the heart because of khat consumption can lead to cardiac remodeling, hypertrophy, and, ultimately, cardiomyopathy. While the study benefits from a large sample size and standardized methodology, its cross-sectional design limits causal inference, as factors like tobacco and alcohol use may have influenced the results. Meanwhile, the focus on an urban population may reduce the generalizability to rural settings, where khat consumption patterns and associated risk factors may differ.

The use of khat can also induce vasoconstriction and platelet aggregation, potentially leading to the development of acute coronary syndrome (ACS) due to reduced blood flow to the heart muscle. While mild khat chewers did not show an increased risk of acute myocardial infarction (AMI), moderate khat chewers were found to have a higher risk (with an odds ratio of 7.62), and heavy khat chewers faced an even greater risk (with an odds ratio of 22.28). In a cohort study conducted by Ali WM et al., khat chewers were more likely to present with ST-Elevation Myocardial Infarction (STEMI) (74.4%), followed by unstable angina (14.3%) and Non-ST-Elevation Myocardial Infarction (NSTEMI) (11.2%; *p* < 0.001). Conversely, non-khat chewers more frequently presented with NSTEMI (34.9%) [14]. Despite the study’s large multicenter cohort design, potential confounders such as delayed healthcare access, pre-existing cardiovascular risk factors, and higher smoking rates among khat users may introduce bias. Furthermore, the mortality rate among khat chewers is higher compared to those who do not chew khat, with rates of 7.5% versus 3.8%, respectively (*p* < 0.001). This trend persisted at 1-month (15.5% versus 6.4%; *p* < 0.001) and 1-year (18.8% versus 10.8%; *p* < 0.001) follow-ups.

Regarding lesion complexity during percutaneous coronary intervention (PCI), there was no significant difference between khat users and non-khat users, as mentioned in the study conducted by Diyar Köprülü et al. [15]. Furthermore, khat chewing is associated with various negative health outcomes, including heart failure, repeated myocardial ischemia or myocardial infarction (MI), ventricular arrhythmia, cardiogenic shock, and stroke. Research conducted by Reyad Mohamed A. Al-sobehi et al. indicated a statistically significant correlation between khat chewing and stroke among individuals under the age of 55 [16]. However, its cross-sectional design did not infer a causal relationship and the presence of confounders such as hypertension, diabetes, and smoking may have influenced the results. Additionally, the selection bias from hospitalized cases reduces its generalizability.

A study conducted on 50 Yemeni patients diagnosed with dilated cardiomyopathy evaluated the impact of habitual khat consumption, in addition to other cardiovascular risk factors. The findings suggested that khat chewing contributes to the development of dilated cardiomyopathy, especially in young patients with a hereditary predisposition [17]. There are several potential mechanisms through which nonischemic cardiomyopathy can occur. The active compounds found in khat may exert direct toxic effects on cardiomyocytes. Prolonged exposure to these compounds could result in cellular damage, impaired contractility, and structural alterations in the myocardium.

Khat consumption has the potential to alter cardiac electrophysiology, which can lead to arrhythmias and conduction abnormalities. Dysregulated ion channels, prolonged QT intervals, and triggered arrhythmias have been reported in association with exposure to cathinone and cathine, which could contribute to the pathogenesis of cardiomyopathy.

A study by Jayed and Al-Huthi (2016) investigated the impact of khat chewing on cardiac rhythm by monitoring 60 individuals on both khat-free and khat-chewing days. The findings revealed a relative increase in ventricular arrhythmias during khat-chewing days [18]. Specifically, non-sustained ventricular arrhythmias (VT) were observed in 23% of cardiac patients on khat-chewing days compared to 6.6% on khat-free days. This was compared to the 3.3% of health controls that experienced non-sustained VT on khat-chewing days, with no occurrences on khat-free days. While the use of 24 h Holter monitoring enhances accuracy, the short observation period may not have captured the long-term risk of arrhythmias. Moreover, the small sample size of this study limits its generalizability.

Khat-induced oxidative stress and inflammation may also contribute to the development of cardiomyopathy (Figure 2). A study published in 2019 [19] revealed that long-term khat chewers demonstrated increased formation of reactive oxygen species (ROS), decreased levels of superoxide dismutase (SOD), and elevated activity of glutathione reductase (GR), indicating heightened oxidative stress and susceptibility to DNA damage and genotoxicity. These factors contribute to myocardial injury and dysfunction.

Genetic factors may also influence individual susceptibility to khat-induced cardiomyopathy. Variations in genes encoding drug-metabolizing enzymes, cardiac ion channels, and signaling pathways involved in cardiac remodeling and repair may impact an individual’s risk of developing cardiomyopathy in response to khat exposure. A study published by Sultan Abdulwadoud Alshoabi et al. demonstrated that khat chewers exhibited a lower mean absolute International Normalized Ratio (INR) reading compared to non-chewers by an average of 0.2 during both initial and subsequent visits (*p* = 0.038 and 0.002, respectively) [20]. The study’s short follow-up period and lack of control for factors like diet, medication adherence, and liver function may have influenced INR levels, making it difficult to determine the effects of khat directly.

Left Ventricular Global Longitudinal Strain (LV GLS) has emerged as a sensitive echocardiographic marker for detecting early myocardial dysfunction in the preclinical stages of khat cardiomyopathy, even before overt systolic impairment becomes evident. LV GLS evaluates the percentage of myocardial deformation during systole, specifically assessing longitudinal fiber shortening from the base to the apex. In cardiomyopathy, myocardial fibrosis, inflammation, or ischemia lead to impaired myocardial fiber contraction, resulting in reduced strain values [21]. Unlike Left Ventricular Ejection Fraction (LVEF), which measures volumetric changes and may remain preserved in early disease stages, LV GLS detects subclinical systolic dysfunction, making it particularly useful for early diagnosis and risk stratification. A reduction in LV GLS (more positive or less negative than the normal range of −18% to −22%) is associated with poor prognosis and increased risk of heart failure, arrhythmia, and mortality [22]. This makes LV GLS a valuable marker in detecting early myocardial dysfunction, guiding clinical decision making, and monitoring disease progression in patients with cardiomyopathy. Khat (Catha edulis) consumption has been associated with myocardial fibrosis, ventricular hypertrophy, and diastolic dysfunction, leading to heart failure. LV GLS also helps differentiate between patients with compensated and decompensated myocardial dysfunction, allowing clinicians to tailor therapeutic strategies accordingly [23]. Additionally, LV GLS has prognostic value in guiding decisions regarding interventions like implantable cardioverter defibrillators (ICDs) and heart failure therapies [24]. By incorporating LV GLS into routine echocardiographic evaluation, clinicians can enhance early detection, risk stratification, and the personalized management of patients with cardiomyopathy.

The cardiovascular impact of khat appears significant, with its stimulant properties contributing to increased blood pressure, arrhythmias, and heightened risks of acute coronary syndromes. Chronic use leads to irreversible cardiac remodeling and heart failure, highlighting the need for further research and early clinical recognition.

## 6. Multiorgan Effects of Khat

Khat’s impact extends beyond the cardiovascular system, affecting various organ systems, including the nervous system (Figure 3). Prolonged and excessive khat use can result in numerous neurological complications. Chronic consumption has been linked to symptoms such as insomnia, anxiety, irritability, and agitation. Moreover, prolonged use may lead to cognitive impairments, including deficits in attention, memory, and executive functioning.

Khat has also been associated with psychiatric disorders, particularly psychosis, especially in susceptible individuals. While khat initially exerts stimulating effects on the nervous system, chronic use can lead to detrimental neurological and psychiatric consequences. It underscores the importance of understanding and addressing the potential risks associated with its consumption.

Animal studies have indicated that compounds found in khat may influence reproductive hormones and sperm quality, resulting in decreased fertility in male animals. In humans, chronic khat use has been associated with various reproductive health issues, including erectile dysfunction and reduced libido in males. Additionally, some studies have reported decreased sperm count and motility in male khat users.

The impact of khat on the gastrointestinal tract can be significant, affecting various aspects of digestive health. The abrasive nature of khat leaves can cause mechanical damage to the oral mucosa, potentially leading to oral ulcers and dental problems. Furthermore, the stimulant properties of cathinone and cathine can enhance gastrointestinal motility, resulting in increased bowel movements and possibly diarrhea.

Moreover, khat consumption has been linked to an elevated risk of gastrointestinal disorders such as gastritis and peptic ulcers. Chronic use of khat has also been implicated in hepatotoxicity, with reported cases of autoimmune hepatitis triggered by its chronic consumption [25].

Along with the cardiovascular system, khat also appears to contribute to neurological, reproductive, and gastrointestinal conditions. These widespread effects reinforce the need for a comprehensive approach to assessing khat-related health risk, particularly in long-term users.

## 7. Discussion

The discussion regarding the inclusion of khat cardiomyopathy (KC) in the differential diagnosis of Non-Ischemic Cardiomyopathy (NICMP) is crucial for healthcare practitioners. The findings from the various studies mentioned above suggest a significant association between chronic khat consumption and the development of cardiomyopathy. Recognizing this association is vital for ensuring timely diagnosis and implementing appropriate management strategies.

The emergence of khat-induced cardiomyopathy (KIC) as a health concern extends beyond the borders of developing countries, notably reaching the United States. This highlights the dynamic interplay of globalization and migration patterns. Historically, KIC was primarily observed in regions where khat chewing was deeply ingrained in cultural practices, particularly across parts of East Africa and the Arabian Peninsula. However, the steady influx of immigrants from these regions to the USA has facilitated the spread of khat consumption practices, subsequently increasing the prevalence of KIC within immigrant communities.

This underscores the importance of healthcare providers being aware of the potential cardiac complications associated with chronic khat use, particularly when treating patients from regions where khat consumption is prevalent. By recognizing KIC as a potential differential diagnosis in NICMP cases, healthcare practitioners can provide more targeted care and interventions for affected individuals, ultimately improving patient outcomes.

The increasing prevalence of khat-induced cardiomyopathy (KIC) in developed nations like the USA raises critical considerations for public health and clinical practice. Firstly, healthcare providers in regions with substantial immigrant populations must remain vigilant in recognizing KIC as a potential cause of cardiomyopathy. This involves considering khat-consumption history and cultural background when evaluating patients presenting with cardiomyopathy symptoms, particularly in immigrant communities where khat use may be prevalent.

Secondly, efforts to address KIC should encompass culturally sensitive healthcare services and targeted public health interventions. Healthcare providers should engage in culturally competent care, understanding and respecting the cultural norms and beliefs surrounding khat use within affected communities. Additionally, public health campaigns should aim to raise awareness about the health risks associated with khat consumption and promote cessation among affected populations. These interventions may include community education programs, outreach initiatives, and support services to assist individuals in reducing or quitting khat use.

By integrating culturally sensitive care practices and targeted public health efforts, healthcare providers and public health authorities can better address the growing prevalence of KIC in developed nations and work towards improving the health outcomes of affected individuals and communities.

Furthermore, there is a pressing need for additional longitudinal studies to deepen our understanding of the direct myocardial effects of khat and its constituents, such as cathinone and cathine. Longitudinal studies would offer valuable insights into the progression and mechanisms of khat cardiomyopathy (KC), shedding light on the pathophysiological pathways involved in khat-induced cardiomyopathy.

Moreover, such longitudinal studies could help identify potential biomarkers for early detection and risk stratification, enabling more precise and targeted therapeutic interventions. By tracking individuals over time, researchers can better assess the impact of chronic khat consumption on cardiac health and identify key factors that contribute to the development and progression of KC.

Overall, longitudinal studies represent a crucial avenue for advancing our understanding of khat-induced cardiomyopathy and developing more effective strategies for the prevention, diagnosis, and management of this condition.

Clinical recommendations should underscore the importance of avoiding khat consumption, especially in individuals with pre-existing cardiovascular risk factors or Non-Ischemic Cardiomyopathy (NICMP). Recognizing and managing khat-related heart problems begins with identifying at-risk patients early. Patients, especially those from endemic areas, can be asked about khat use, including how often and how much is consumed, especially in patients with unexplained heart issues like arrhythmia, high blood pressure, or cardiomyopathy. Identifying these complications early allows for better targeted treatment and can help prevent more serious heart problems down the line. Healthcare providers should prioritize educating patients about the potential cardiac complications associated with khat use and should actively encourage cessation of this habit.

Moreover, for individuals diagnosed with khat cardiomyopathy (KC) or NICMP secondary to khat consumption, comprehensive management strategies, including guideline-directed medical therapy (GDMT), should be promptly implemented. Although the extent of reversibility of cardiac dysfunction in these patients remains uncertain, early cessation of khat chewing combined with optimal medical management may lead to improvements in left ventricular ejection fraction (LVEF) and overall prognosis.

Further research is warranted to explore the potential for LVEF normalization, with or without GDMT, in individuals who abstain from khat use. Additionally, ongoing studies should investigate the long-term outcomes and optimal treatment approaches for individuals with KC or NICMP associated with khat consumption, ultimately guiding clinical practice and improving patient care.

## 8. Conclusions

In conclusion, recognizing khat cardiomyopathy (KC) as a potential differential diagnosis of Non-Ischemic Cardiomyopathy (NICMP), conducting longitudinal studies to assess direct myocardial effects, and providing clinical recommendations for khat avoidance and potential left ventricular ejection fraction (LVEF) recovery are essential steps in addressing the cardiovascular implications of khat consumption and enhancing patient outcomes. Overall, addressing the cardiovascular implications of khat consumption requires a multifaceted approach that includes clinical vigilance, research advancements, and public health initiatives aimed at promoting awareness and cessation of this harmful habit.

## Figures and Tables

**Figure 1 jcm-14-01100-f001:**
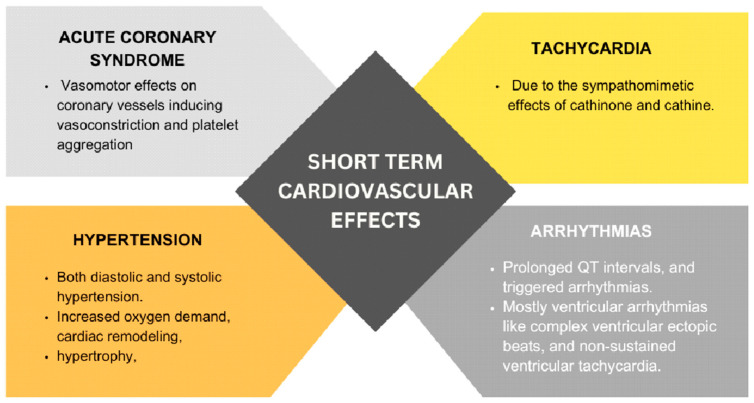
Short-term cardiovascular effects of khat.

**Figure 2 jcm-14-01100-f002:**
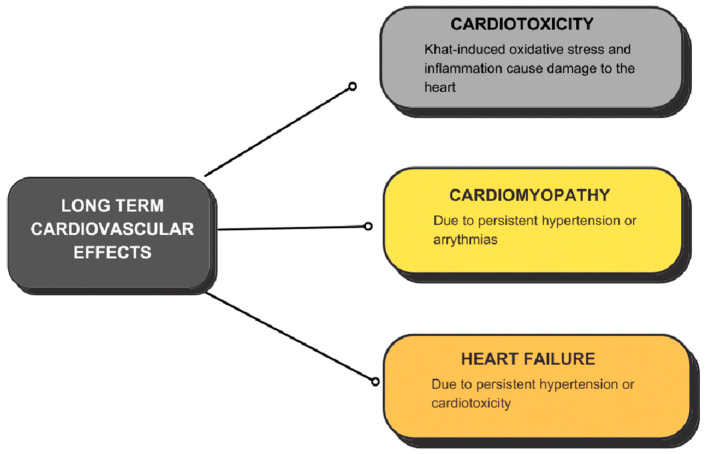
Long-term cardiovascular effects of khat.

**Figure 3 jcm-14-01100-f003:**
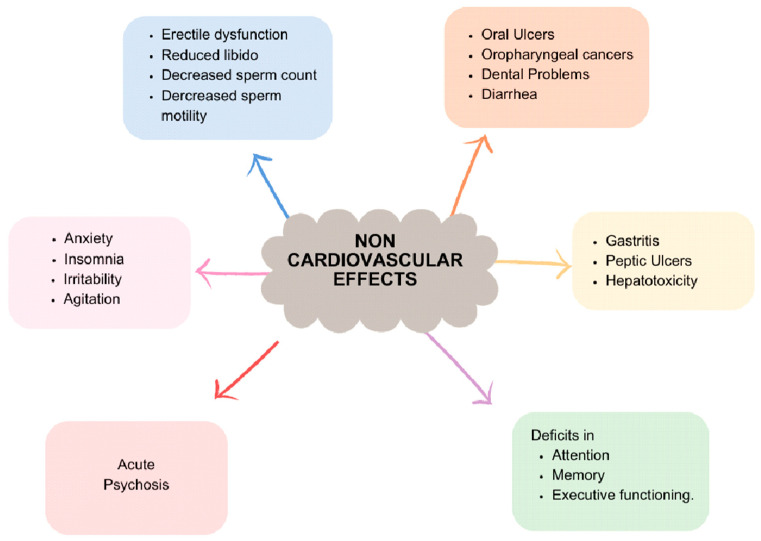
Non-cardiovascular effects of khat.

## Data Availability

No new data were created.

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
