# Peer review of "Understanding the Intricacies of Khat-Associated Cardiovascular Disease"

_jcm, 2025, doi:10.3390/jcm14041100_

Round 1
Reviewer 1 Report
Comments and Suggestions for Authors
In this interesting paper, the authors focused on Khat, its medical and non-medical uses, its cardiovasscular and non-cardiovascular effects.
Its psychoactive compounds, including cathinone and cathine, exert sympathomimetic effects by inducing the release of catecholamines such as adrenaline and noradrenaline.
The short-term cardiovascular effects of khat are tachycardia, ventricular arrhythmias, hypertension and acute coronary syndromes.
The long-term cardiovascular effects of khat are cardiotoxicity, cardiomyopathy and heart failure.
The authors also listed several noncardiovascular effects of Khat, particularly psychiatric disorders.
The manuscript is well written and very interesting for clinical cardiologists.
I have a few suggestions for the authors.
On Page 8, the title "The Conclusion" could be replaced by "Discussion".
The authors could expand the paragraph concerning the Khat cardiomyopathy, by discussing also the potential subclinical findings of this disease.
Indeed, the authors stated that, to date, longitudinal studies are needed for identifying potential biomarkers for early detection and risk stratification of Khat cardiomyopathy.
With this regards, on line 301, the authors could also discuss the potential usefulness of speckle tracking echocardiography (STE), an innovative imaging modality, for early detecting subclinical myocardial dysfunction, defined as the impairment in left ventricular (LV) global longitudinal strain (GLS) in the presence of preserved left ventricular ejection fraction (LVEF). The authors could cite and discuss the following references: PMID: 29889664, PMID: 23563128 and PMID: 32519318. Further studies assessing LV-GLS by STE in the preclinical stages of Khat cardiomyopathy might improve its early detection, medical management and prognostic risk stratification.
Author Response
Comment 1: On Page 8, the title "The Conclusion" could be replaced by "Discussion".
Response 1: Changed
Comment 2: The authors could expand the paragraph concerning Khat cardiomyopathy, by discussing also the potential subclinical findings of this disease.
Indeed, the authors stated that, to date, longitudinal studies are needed for identifying potential biomarkers for early detection and risk stratification of Khat cardiomyopathy.
With this regards, on line 301, the authors could also discuss the potential usefulness of speckle tracking echocardiography (STE), an innovative imaging modality, for early detecting subclinical myocardial dysfunction, defined as the impairment in left ventricular (LV) global longitudinal strain (GLS) in the presence of preserved left ventricular ejection fraction (LVEF). The authors could cite and discuss the following references: PMID: 29889664, PMID: 23563128 and PMID: 32519318. Further studies assessing LV-GLS by STE in the preclinical stages of Khat cardiomyopathy might improve its early detection, medical management and prognostic risk stratification.
Response 2: Comments addressed
Left Ventricular Global Longitudinal Strain (LV GLS) has emerged as a sensitive echocardiographic marker for detecting early myocardial dysfunction in the preclinical stages of khat cardiomyopathy, even before overt systolic impairment becomes evident. LV GLS evaluates the percentage of myocardial deformation during systole, specifically assessing longitudinal fiber shortening from the base to the apex. In cardiomyopathy, myocardial fibrosis, inflammation, or ischemia lead to impaired myocardial fiber contraction, resulting in reduced strain values [21]. Unlike LVEF, which measures volumetric changes and may remain preserved in early disease stages, LV GLS detects subclinical systolic dysfunction, making it particularly useful for early diagnosis and risk stratification. A reduction in LV GLS (more positive or less negative than the normal range of -18% to -22%) is associated with poor prognosis and increased risk of heart failure, arrhythmias, and mortality[22]. This makes LV GLS a valuable marker in detecting early myocardial dysfunction, guiding clinical decision-making, and monitoring disease progression in patients with cardiomyopathy. Khat (Catha edulis) consumption has been associated with myocardial fibrosis, ventricular hypertrophy, and diastolic dysfunction, leading to heart failure. LV GLS also helps differentiate between patients with compensated and decompensated myocardial dysfunction, allowing clinicians to tailor therapeutic strategies accordingly [23]. Additionally, LV GLS has prognostic value in guiding decisions regarding interventions like implantable cardioverter defibrillators (ICDs) and heart failure therapies [24]. By incorporating LV GLS into routine echocardiographic evaluation, clinicians can enhance early detection, risk stratification, and personalized management of patients with cardiomyopathy.
References:
21. Sonaglioni A, Albini A, Fossile E, Pessi MA, Nicolosi GL, Lombardo M, Anzà C, Ambrosio G. Speckle-Tracking Echocardiography for Cardioncological Evaluation in Bevacizumab-Treated Colorectal Cancer Patients. Cardiovasc Toxicol. 2020 Dec;20(6):581-592. doi: 10.1007/s12012-020-09583-5. PMID: 32519318.
- Kaufmann D, Szwoch M, Kwiatkowska J, Raczak G, Daniłowicz-Szymanowicz L (2019) Global longitudinal strain can predict heart failure exacerbation in stable outpatients with ischemic left ventricular systolic dysfunction. PLoS ONE 14(12): e0225829. https://doi.org/10.1371/journal.pone.0225829
- Jung, I. H., Park, J. H., Lee, A., Kim, G. S., Lee, H. Y., Byun, Y. S., & Kim, B. O. (2020). Left Ventricular Global Longitudinal Strain as a Predictor for Left Ventricular Reverse Remodeling in Dilated Cardiomyopathy. Journal of Cardiovascular Imaging, 28(2), 137. https://doi.org/10.4250/jcvi.2019.0111
- Ng ACT, Delgado V, Bax JJ. Application of left ventricular strain in patients with aortic and mitral valve disease. Curr Opin Cardiol. 2018 Sep;33(5):470-478. doi: 10.1097/HCO.0000000000000538. PMID: 29889664.

Reviewer 2 Report
Comments and Suggestions for Authors
The manuscript provides an in-depth review of the cardiovascular implications of khat use, supported by a comprehensive analysis of existing literature. The inclusion of socio-cultural factors and legal status of khat adds depth to the discussion. However, there are areas where the manuscript can be improved.
The abstract provides a good overview but could be more structured. Include a brief mention of the recommendations and gaps in research.
The objectives of the manuscript are stated, but they could benefit from clearer articulation in the introduction. Define the scope of the review more explicitly.
The manuscript transitions well between sections but lacks concise summaries at the end of each subsection (e.g., cardiovascular effects, multiorgan effects). Adding a short synthesis at the end of each section would improve readability.
The manuscript does an nice job of summarizing studies, but it lacks critical analysis of the strengths, weaknesses, and limitations of the cited research. For instance, comment on the sample sizes, potential biases, or regional limitations of studies.
The manuscript mentions cardiac arrhythmias as a potential consequence of khat consumption, referencing studies that highlight the role of cathinone and cathine in altering cardiac electrophysiology. However, it would be helpful if the authors could elaborate on whether there are detailed data available on the frequency, severity, and types of arrhythmic events (e.g., atrial fibrillation, ventricular tachycardia) observed in khat users. Are there specific studies or clinical cases that quantify these arrhythmic events?
While recommendations for public health initiatives are mentioned, more specific and actionable clinical recommendations for managing patients with khat-induced cardiovascular conditions would be beneficial.
Author Response
Comment 1: The manuscript transitions well between sections but lacks concise summaries at the end of each subsection (e.g., cardiovascular effects, multiorgan effects). Adding a short synthesis at the end of each section would improve readability.
Response 1: The brief summary is added at the end of each section
Comment 2: The manuscript does an nice job of summarizing studies, but it lacks critical analysis of the strengths, weaknesses, and limitations of the cited research. For instance, comment on the sample sizes, potential biases, or regional limitations of studies.
Response 2: The critical analysis is added for the cited research , mentioned in the updated manuscript
Comment 3: The manuscript mentions cardiac arrhythmias as a potential consequence of khat consumption, referencing studies that highlight the role of cathinone and cathine in altering cardiac electrophysiology. However, it would be helpful if the authors could elaborate on whether there are detailed data available on the frequency, severity, and types of arrhythmic events (e.g., atrial fibrillation, ventricular tachycardia) observed in khat users. Are there specific studies or clinical cases that quantify these arrhythmic events?
Response 3: Following paragraph added.
A study by Jayed and Al-Huthi (2016) investigated the impact of khat chewing on cardiac rhythm by monitoring 60 individuals on both khat-free and khat-chewing days. The findings revealed a relative increase in ventricular arrhythmias during khat-chewing days. Specifically, non-sustained ventricular arrhythmias (VT) observed in 23% of cardiac patient on khat chewing days compared to 6.6% on khat-free days. This was compared to the 3.3% of health controls that experienced non-sustained VT on khat chewing days, with no occurrences on khat-free days. While the use of 24-hour holter monitoring enhances accuracy, the short observation period may not have captured long-term risk of arrhythmias. Moreover, the small sample size of this study limits generalizability.
Comment 4: While recommendations for public health initiatives are mentioned, more specific and actionable clinical recommendations for managing patients with khat-induced cardiovascular conditions would be beneficial.
Response 4: Recognizing and managing khat-related heart problems begin with identifying at-risk patient early. Patients, especially those from endemic areas, can be asked about khat use, including how often and how much is consumed, especially in patients with unexplained heart issues like arrhythmias, high blood pressure, or cardiomyopathy. Identifying these complications early allows for better-targeted treatment and can help prevent more serious heart problems down the line

Round 2
Reviewer 2 Report
Comments and Suggestions for Authors
I have no further comments.